# COVID-19 Vaccination in Romania and the Benefits of the National Electronic Registry of Vaccinations

**DOI:** 10.3390/vaccines11020370

**Published:** 2023-02-06

**Authors:** Bianca Georgiana Enciu, Daniela Pițigoi, Alina Zaharia, Rodica Popescu, Andreea Niculcea, Maria-Dorina Crăciun, Adriana Pistol

**Affiliations:** 1Department of Epidemiology, Carol Davila University of Medicine and Pharmacy, 020021 Bucharest, Romania; 2National Centre for Communicable Diseases Surveillance and Control, National Institute of Public Health, 050463 Bucharest, Romania; 3Ministry of Health, 010024 Bucharest, Romania

**Keywords:** vaccination registry, COVID-19, Romania

## Abstract

Background: Recording real-time data of vaccinations performed, vaccine stocks, and adverse events following immunization is a particularly useful activity in the effective development of any vaccination campaign or vaccination program, guiding the decisions of public health authorities. The aim of this paper is to present the benefits of the National Electronic Registry of Vaccinations in providing useful information for the optimization of healthcare vaccination policies, specifically related to COVID-19 vaccination. Methods: We performed a descriptive study using data available in the reports generated from the National Electronic Registry of Vaccinations regarding COVID-19 vaccinations performed between 27 December 2020 and 31 December 2021. Results: A total of 27,980,270 doses of the COVID-19 vaccine were distributed. Of these, 15,757,638 (56%) were administered in 4545 vaccination centers: 7,882,458 as the 1st dose (50%), 5,878,698 as the 2nd dose (37%), and 1,996,482 as the 3rd dose (13%). More than 25% of the total doses were administered to people over 65 years of age. A total of 41% of the population received at least one dose of the COVID-19 vaccine. A total of 4083 adverse events following immunization were reported. Conclusions: The existence of a National Electronic Registry of Vaccinations containing accurate information on vaccinations performed in Romania offers the opportunity to obtain a clear picture of vaccination status that will significantly contribute to the optimization of vaccination strategies and programs.

## 1. Introduction

Vaccination is a complex public health intervention which has major health benefits. The COVID-19 pandemic has reminded everyone of the importance of vaccines in the prevention and control of diseases [1].

As soon as the first COVID-19 vaccines were available, efforts were directed towards the extensive production, purchase, and distribution of the vaccines, following which large-scale vaccination was pursued in order to reduce the morbidity, mortality, and economic impact of COVID-19.

The European Union has facilitated COVID-19 vaccine acquisition for Member States. However, the results of the vaccination campaigns differ from country to country, being dependent on the economic level, organization, and financing of the health system, as well as on the population’s level of trust in the vaccine and in their authorities [2,3]. The availability of and access to COVID-19 vaccines were limited in the initial phases of the vaccination campaign. For this reason, the World Health Organization (WHO) and the European Centre for Disease Control and Prevention (ECDC) issued a series of recommendations to prioritize vaccination among certain at-risk groups. Thus, in Romania, as in other European countries, vaccination was implemented in phases: (1) health and social services workers (physicians, nurses, paramedics, volunteers, students, resident doctors, and social workers), (2) at-risk populations (people with disabilities and chronic diseases, people over 65 years of age, and people experiencing homelessness) and those employed in essential sectors, (3) the general population (including the pediatric population, depending on the epidemiological evolution and the characteristics of vaccines approved for use in people under the age of 18) [2,3,4].

When defining these population groups, multiple factors were taken into consideration, such as: social ethics and equity principles, epidemiologic criteria, medical criteria (risk of infection, risk of severe disease and death, risk of further transmission), as well as their impact on essential activities ensuring the proper functioning of critical infrastructure [4]. Furthermore, depending on medical criteria, epidemiologic trends, indications and contraindications for the types of vaccines approved, and the availability of different vaccines, several sub-categories were prioritized within each category [4].

The effective implementation of the COVID-19 vaccination campaigns at the national level is also dependent on the ability to access, receive, and transmit information as quickly as possible between management and staff at all levels. The existence or development of electronic platforms is a major necessity when it comes to monitoring vaccination coverage and inventory management (which promotes sustainable supply and avoids losses).

In Romania, only the COVID-19 vaccines approved by the European Medicine Agency (EMA) were used. The eligible population groups and the vaccination schedule were both updated according to international recommendations. The vaccination is voluntary and free of charge for the entire population. The cost of the vaccination is supported by the health authorities.

Online reporting of vaccinations to the National Electronic Registry of Vaccinations has been practiced since 2011 by family doctors and from maternities; this was initially utilized for vaccines administered to children and adolescents according to the National Vaccination Schedule [5,6,7].

In 2020, before the beginning of the COVID-19 vaccination campaign, a section for the adult population, dedicated to COVID-19 vaccination, was added to the National Electronic Registry of Vaccinations, which allows for the recording of data regarding the vaccinated person (identification data and risk group according to the COVID-19 vaccination strategy), data regarding the vaccine administered (type, brand, vaccine batch number, and expiration date), and data regarding adverse events following immunization.

A digital vaccination certificate downloaded from the National Electronic Registry of Vaccinations is provided to each vaccinated person containing the essential information regarding vaccination (date, place, number of doses, vaccine details). Additionally, this new section includes information regarding the working procedures for healthcare professionals developed by the National Coordination Committee for COVID-19 Vaccination Activities (CNCAV) and the National Electronic Registry of Vaccinations technical support team.

The aim of this paper is to present the benefits of the National Electronic Registry of Vaccinations in providing useful information for the optimization of the COVID-19 vaccination process.

## 2. Materials and Methods

We performed a descriptive study regarding the COVID-19 vaccination campaign in Romania using the reports generated from the National Electronic Registry of Vaccinations. The National Electronic Registry of Vaccinations is an electronic application used for the online reporting of vaccinations performed in Romania. It is a technical database of vaccinations of children, adolescents, and adults from Romania based on individual records. The National Centre for Communicable Diseases Surveillance and Control is responsible for the National Electronic Registry of Vaccinations and the authors have contributed to the development, implementation, and management of this database. Personal data confidentiality is ensured according to the current legal framework [4,8,9]. We generated the reports on 2 March 2022. The selected study period was from 27 December 2020 to 31 December 2021. The following data were extracted: number of doses distributed (total, by vaccine brand), number of doses administered (total, by vaccine brand and by sex, age, and county), number of doses lost (total, by vaccine brand and by cause), and the total number of adverse events following immunization reported in the National Electronic Registry of Vaccinations.

In Romania, the adverse events following immunization are passively recorded in the National Electronic Registry of Vaccinations by the vaccinating physicians or reported by filling in the report form and an event investigation report (only for serious adverse events) to be passed on to the National Centre for Communicable Diseases Surveillance and Control. Additionally, the pharmacovigilance system managed by the National Agency for Medicines can be used to make an online report of an adverse event [10,11,12].

Data regarding the Romanian population on 1st January 2021 available on the National Institute of Statistics website were used for estimating the number of doses administered per 100 inhabitants and the regional vaccine uptake rate [13].

All data were processed according to the current legal framework regarding the confidentiality of personal data. For data analysis, we used Microsoft Excel 365.

## 3. Results

The COVID-19 vaccination campaign started on 27 December 2020 (with the vaccination of medical staff). During the study period, this campaign took place in 4545 vaccination centers: 1500 specially organized for COVID-19 vaccination and 3045 family medicine offices. Additionally, in order to increase access to vaccination among people from vulnerable groups, mobile vaccination teams were organized.

The main characteristics of the COVID-19 vaccines used in Romania are presented in Table 1.

In total, 27,980,270 doses of the COVID-19 vaccine were distributed by the Ministry of Health during the study period. Of these, 15,757,638 doses (56%) were administered and 700,735 (3%) doses were lost. Additionally, 4,779,830 doses were donated by the Ministry of Health to neighboring countries with poor access to COVID-19 vaccines (e.g. Republic of Moldova).

The first available vaccine was the Comirnaty/Pfizer–BioNTech vaccine. The other three vaccines became available in Romania in the following order: Spikevax/Moderna (first used in Romania on 4 February 2021), Oxford–AstraZeneca (first used in Romania on 15 February 2021), and Janssen/Johnson&Johnson (first used in Romania on 4 May 2021).

The Comirnaty/Pfizer–BioNTech vaccine represented the largest quantity of available vaccine doses (17,363,970 doses distributed; 62%); 69% of the distributed doses were used during the study period (Figure 1).

The Comirnaty/Pfizer–BioNTech vaccine was most commonly used as a first dose (5,143,066 doses; 65%), followed by the Janssen/Johnson&Johnson vaccine (1,903,921 doses; 24%). For the second dose, the Comirnaty/Pfizer–BioNTech vaccine (5,049,245 doses; 86%) was the most used, followed by the Vaxzevria/AstraZeneca vaccine (417,859 doses; 7%). Additionally, the Comirnaty/Pfizer–BioNTech vaccine was used in over 90% of people who received a third dose (1,842,847 doses; 92%) (Table 2).

The median number of doses administered weekly was 265,140 (IQR = 333,360.75) with a maximum of 816,798 doses administered during the week of 25 October 2021–31 October 2021; most of these were the first dose in the series (575,434; 70%). The majority of the doses administered as the third dose were administered during the week of 04 October 2021–10 October 2021 (194,838; 10%) (Figure 2).

The pattern of vaccine administration over time was influenced by a number of factors, such as the availability of the vaccine, the population’s perception of the vaccine and the severity of the disease, and the public health measures imposed by authorities. In Figure 3, we can see that at the very beginning of the vaccination campaign, all of the vaccines available at that time were highly utilized. The use of the Vaxzevria/AstraZeneca vaccine fell sharply by the second half of March 2021 as a consequence of the concerns related to the safety of this vaccine. During the summer of 2021, the number of doses administered (regardless of the brand) was low because the severity of the pandemic decreased during that period and the authorities mitigated the public health measures. At the end of September 2021, the administration of the booster dose began, with mRNA vaccines being mainly recommended as a booster dose regardless of the vaccine used as the primary vaccination.

In addition, the implementation of mandatory presentation of the digital vaccination certificate in order to have access to certain public and private institutions explains the high number of doses administered during the fall of 2021, especially during October 2021. The Janssen/Johnson&Johnson vaccine was the most commonly utilized vaccine during that period as the primary vaccination. People opted for this product because its one dose primary vaccination offers the opportunity to obtain the digital vaccination certificate more quickly (at 10 days after its administration) (Figure 3).

The Comirnaty/Pfizer–BioNTech vaccine accounted for the highest number of lost doses (426,866; 61% of the total losses; 2.5% of the total doses distributed) (Table 3).

The most common causes of loss were defrosting and dilution and subsequent non-utilization, as the COVID-19 vaccines are conditioned in multi-dose vials and have limited stability after dilution (Figure 4).

During the first quarter of the vaccination campaign, dilution and non-utilization and the interruption of the cold chain (31% and 29%, respectively) were the main causes of vaccine loss. In comparison, during the last two quarters of 2021, slaughter for the benefit of vaccination represented a significant proportion of the total losses (29% during the 3rd quarter and 14% during the 4th quarter) (Figure 5).

Regarding the causes of loss by vaccine brand, it is noted that in the case of the Vaxzevria/AstraZeneca vaccine, most doses were lost because of expiration compared to the other vaccines, which were mostly lost due to expiration following defrosting and dilution and non-utilization.

Out of 15,757,638 administered doses of the COVID-19 vaccine, 7,882,458 were administered as the 1st dose (50%), 5,878,698 as the 2nd dose (37%), and 1,996,482 as the 3rd dose (13%). The rate of vaccination uptake reached 41% during this time period.

Regarding the distribution of the doses administered by gender, a slight male preponderance was observed (8,217,068; 52%). The vaccination uptake rate for at least one dose of vaccine was 40% for women and 43% for men.

By age group, more than 25% of the total number of doses were administered to people over 65 years of age (4,085,094; 26%), with 1,768,762 (22%) vaccines administered as the 1st dose, 1,520,496 (26%) as the 2nd dose, and 795,836 (40%) as the 3rd dose (Figure 6).

In total, 861,125 doses (5.4% overall) were administered to healthcare workers, including those involved in the management of the COVID-19 pandemic. Thus, 341,001 healthcare workers received at least one dose of the COVID-19 vaccine, and the vaccination uptake rate among this group stands at 94%.

Moreover, 99,274 doses were used to vaccinate people from institutional centers or those experiencing homelessness (0.6%).

The vaccination uptake rate varies from county to county. Romania is divided into 41 counties and Bucharest, the capital city. Most doses were administered in Bucharest (2,271,376; 14%). The fewest number of doses/100 inhabitants was recorded in Suceava (40/100 inhabitants).

The vaccination coverage by administrative territoral unit, meaning locality, city or county, is presented in Figure 7.

In addition to this, 59,946 people were vaccinated while in Romania but with a passport for another country or were Romanian but vaccinated abroad. For people vaccinated abroad, the National Registry of Vaccinations allowed the introduction of doses administered abroad in order to provide a complete vaccination history.

A total of 4083 adverse events following immunization were reported (2718 for Comirnaty/Pfizer–BioNTech, 785 for Vaxzevria/AstraZeneca, 444 for Janssen/Johnson &Johnson, and 135 for Spikevax/Moderna). The number of adverse events following immunization at 1000 doses administered did not differ substantially between the four types of vaccine products.

## 4. Discussion

Since the beginning of the COVID-19 vaccination campaign in Romania, the National Electronic Registry of Vaccinations has provided the information needed to facilitate the development of the vaccination process (real-time stock management and vaccine supply accordingly, data on vaccine losses, and data on vaccination coverage by population groups or territorial administrative unit). A portion of this information has been transparently communicated to the Romanian public (on TV, social media, and other websites).

In this research, we set out to highlight the role of the National Electronic Registry of Vaccinations in providing useful information for the optimization of healthcare vaccination policies. To achieve this goal, we present a summary of the first year of COVID-19 vaccination in Romania using data provided by the National Electronic Registry of Vaccinations.

In Romania, the vaccination campaign started simultaneously with the COVID-19 vaccination campaigns in most European countries. However, Romania is ranked last among European countries in terms of vaccination coverage against COVID-19, and the weekly evolution of the vaccination campaign reflects not only the availability of the COVID-19 vaccines (which were available in limited quantities in the initial stages of the vaccination campaigns due to insufficient production capacity and increased demand), but also the population’s perception of the benefits of vaccination and the risk of contracting the disease and the impact of the measures imposed by the authorities to limit the spread of SARS-CoV-2. Therefore, there was a decrease in interest in vaccination from May 2021 onward, due to the mitigation of some public health measures, and a subsequent increase in interest after the imposition of mandatory presentation of the digital vaccination certificate in order to access certain public and private institutions [14]. This variation suggests a need to update the legal framework to support the increase in vaccination coverage.

Moreover, the number of doses of different brands of vaccines administered during this time was influenced by their differing vaccination schedules. For example, the administration of booster doses after primary vaccination began at the end of September 2021, and at that time mRNA vaccines were recommended for use as booster doses. The Vaxzevria/Oxford–AstraZeneca vaccine was not recommended as a booster dose. Additionally, the population’s perception of one product could influence the acceptance of that brand for vaccination.

The vaccination uptake rate varies from county to county. For example, Suceava, a county that faced great problems with managing the COVID-19 pandemic at its inception, recorded the lowest rate of vaccination uptake. This fact suggests the need to improve public health strategies at the local/county level [15]. The data on distribution by county have certain limitations related to the fact that some people may have their official address in one county but reside in another.

Regarding the distribution by age groups, we noticed that the highest number of doses was used to vaccinate people over 65 years of age. However, within this age group, a lower number of doses was used to vaccinate people over 85 years of age, a finding which could be explained by the difficulties experienced by this group in accessing health services. Efforts have been made to increase this group’s access to vaccines (for example, the organization of mobile vaccination teams). Additionally, a small number of doses was used to vaccinate people under 20 years of age, suggesting increased reluctance to vaccination among young people or parents, in the case of minors.

About 4000 adverse events following immunization were reported by vaccinating physicians to the National Electronic Registry of Vaccinations at about 16 million doses administered. To gain a clear picture of the scope and nature of adverse events following immunization, it is necessary to increase the interest of medical staff in reporting them (both vaccinating medical staff and family doctors). Additionally, to reduce the vaccination hesitancy caused by the fear of possible adverse reactions, the data collected should be analyzed by a multidisciplinary team of experts and the results should be transparently communicated to those who need this information [1].

At the beginning of the vaccination campaign, vulnerabilities related to vaccine storage and the distribution chain were identified as challenges in achieving proper progress in the COVID-19 vaccination campaign in Romania [2]. However, during the first year of the COVID-19 vaccination campaign, vaccine losses amounted to less than 3% of the total number of doses distributed. The main reasons for losses were defrosting, dilution and non-utilization, interruption of the cold chain, slaughter for the benefit of vaccination, and expiration. The causes of these losses are acceptable given the fact that the COVID-19 vaccines are conditioned in multi-dose vials (in the case of the Spikevax vaccine, the booster vaccination was composed of half a dose) with limited stability after dilution. To reduce these vaccine losses at the beginning of the vaccination campaign when the availability of vaccines was limited, the people who wanted to get vaccinated were required to schedule their vaccination in advance. As the availability of vaccines increased, appointments were no longer required in order to achieve higher vaccination coverage. This low percentage of losses demonstrates the benefit of availability of real-time data on problems arising along the supply chain, which facilitates rapid intervention. However, identifying that interruptions in the cold chain were the third most prevalent cause of losses suggests the need to improve it.

The COVID-19 vaccination campaign represented a major challenge for Romania, given its status as a country with a low GDP per capita allocated to health, with an increased mortality rate because of treatable conditions and a low vaccination rate [2]. However, through proper coordination, involvement of medical workers, local, and central authorities, and real-time monitoring of the process, an appropriate response to the epidemiological situation was organized.

In our opinion, the vaccination of over 40% of the eligible population against COVID-19 is a reasonable achievement, given the reluctance of the Romanian population towards preventive measures. This achievement is important because it was accomplished with recently licensed vaccines. The vaccination uptake rate among medical workers is almost double the vaccination uptake rate among the general population. This is indicative of the responsible attitude of medical personnel and their high level of awareness regarding the severity of the disease and the importance of the COVID-19 vaccination in preventing the disease as well as the effective organization of the COVID-19 vaccination campaign (in terms of vaccination prioritization and the availability of vaccines in sufficient quantities).

However, the data presented above suggest the need to implement additional measures to increase interest in vaccination, to improve the vaccine distribution process, and to improve reporting of adverse events.

The starting point for these is the National Electronic Registry of Vaccinations, which has been developed and permanently adapted to the needs of the vaccination campaign and which has managed to fulfill the roles that an electronic registry created for monitoring vaccinations should perform [1,16]: to monitor vaccine stocks, administration, and the safety of vaccines. Additionally, it allows to provide information to health professionals for the unified implementation of the vaccination process and vaccination certificates for vaccinated people.

Moreover, the Special Telecommunications Service (STS) has connected the National Electronic Registry of Vaccinations with other electronic platforms, such as the vaccination scheduling platform (which allowed the retrieval of user data from the National Electronic Registry of Vaccination), the Corona.Forms platform (which allowed the retrieval of vaccination history for tested/confirmed/deceased persons), the Romanian Civil Registry of People (which allowed the validation of identity data), the Health Insurance House (for settling the vaccination service fees for family doctors), and the platform used for issuing the European digital certificate of vaccination.

The integration with the Corona.Forms platform is very important, because it allows for the estimation of vaccine effectiveness.

Starting in July 2022, a unique form similar to that created for adult vaccinations was implemented for reporting all vaccinations (of children, adolescents, and adults) given the advantages proved by this model during the 18 months of the COVID-19 vaccination campaign.

These new developments in the National Electronic Registry of Vaccinations were followed by user training sessions to ensure the accuracy and completeness of the information recorded.

This study has a strengths in offering an image of the usefulness of vaccination registries in providing information for improving vaccination strategies (vaccination coverage, age groups reluctant to receive vaccinations, county, and regions with deficiencies in implementing the vaccination program).

The National Electronic Registry of Vaccinations is a database with a high level of completeness and accuracy. Data regarding COVID-19 vaccination have proven their correctness over time even though the accuracy and completeness of the data is dependent on the users who record the vaccinations.

Limitations: The limitations of this study are related to the insufficient data available to describe adverse events following COVID-19 immunization and the distribution of administered doses by risk category.

## 5. Conclusions

In Romania, the COVID-19 vaccines were available immediately after their authorization in Europe in sufficient quantities. However, the vaccination coverage recorded during the studied period is one of the lowest in the region. The National Electronic Registry of Vaccinations allowed real-time monitoring of vaccinations, consumption, vaccine stocks, and immediate adverse events following immunization. Recording accurate information on vaccinations performed at the national level offers a clear picture of the vaccination campaign and can significantly contribute to the optimization of vaccination strategies and programs.

## Figures and Tables

**Figure 1 vaccines-11-00370-f001:**
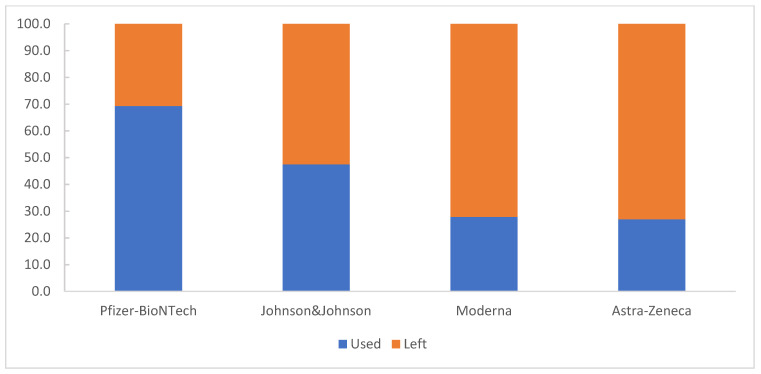
The utilization rates of vaccine doses, by type of product, Romania, 27 December 2020–31 December 2021.

**Figure 2 vaccines-11-00370-f002:**
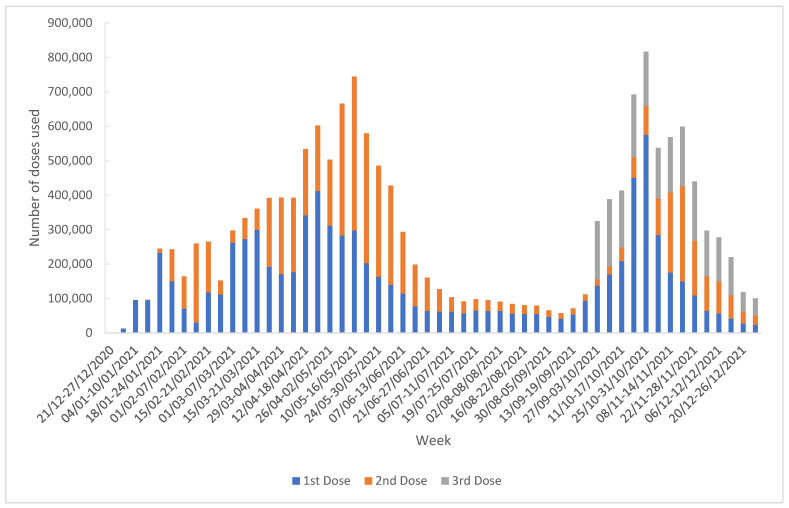
The weekly evolution of the COVID-19 vaccination campaign, Romania, 27 December 2020–31 December 2021.

**Figure 3 vaccines-11-00370-f003:**
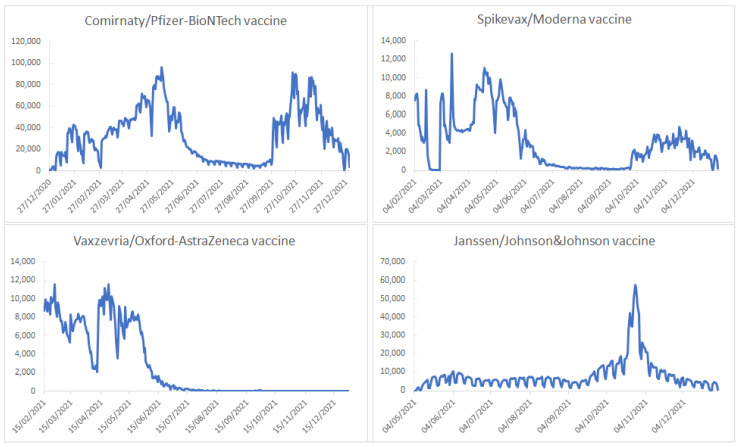
The pattern of vaccine use by brand over time, Romania, 27 December 2020–31 December 2021.

**Figure 4 vaccines-11-00370-f004:**
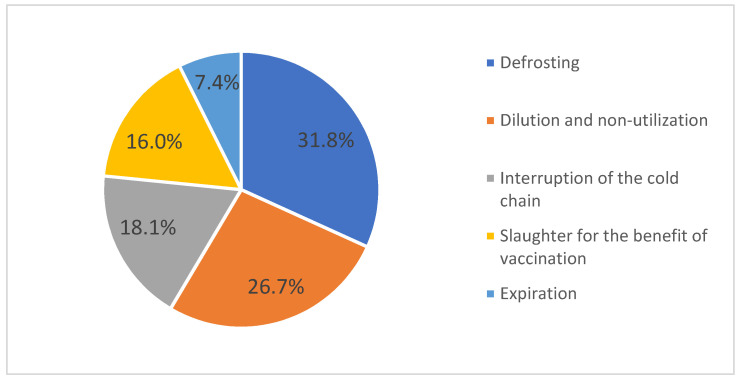
Reported causes of COVID-19 vaccine losses, Romania, 27 December 2020–31 December 2021.

**Figure 5 vaccines-11-00370-f005:**
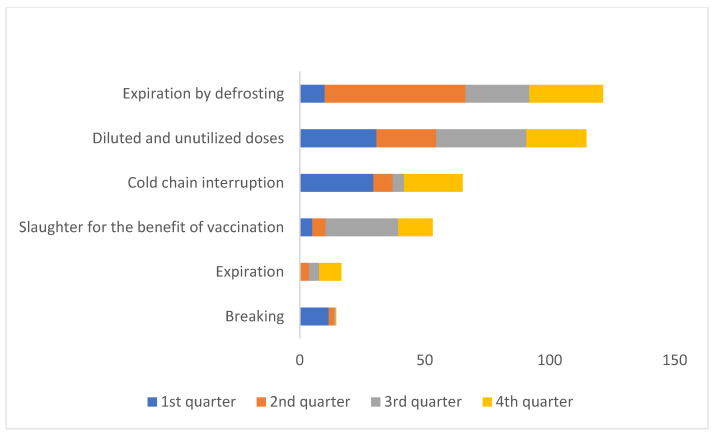
Quarterly distribution of COVID-19 vaccine losses by cause, Romania, 27 December 2020–31 December 2021.

**Figure 6 vaccines-11-00370-f006:**
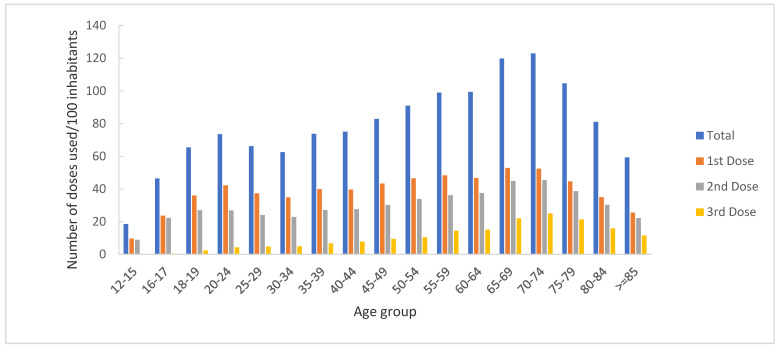
Distribution of administered doses by age group, Romania, 27 December 2020–31 December 2021.

**Figure 7 vaccines-11-00370-f007:**
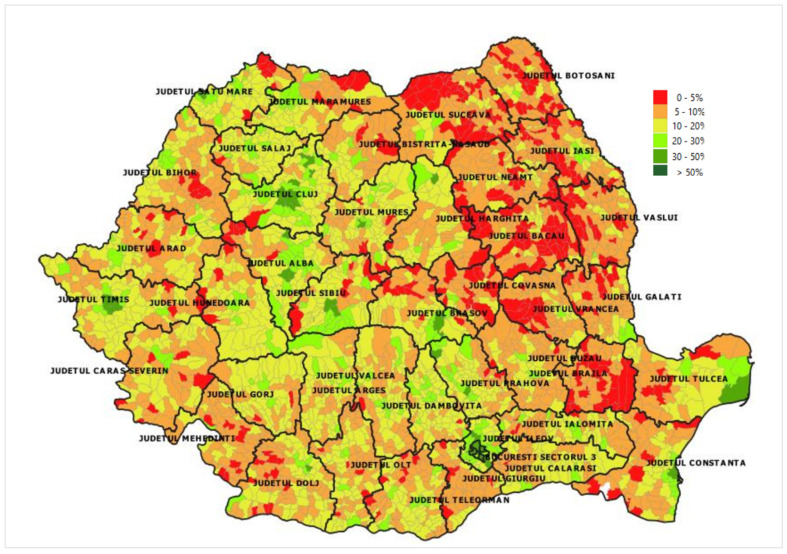
COVID-19 vaccination coverage (at least one dose) in Romania by administrative territorial unit on 31 May 2021, data generated from the National Electronic Registry of Vaccinations.

**Table 1 vaccines-11-00370-t001:** Vaccines used in Romania, 27 December 2020–31 December 2021.

Product (Authorization Date by the European Medicine Agency)	Type of Vaccine	Vaccination Schedule	Indication
Comirnaty/Pfizer–BioNTech (21 December 2020)	mRNA	2 doses at 21 days apart *,**	People aged 12 and over (for the 5–12-year-old group, the pediatric vaccine became available in January 2022)
Spikevax/Moderna (06 January 2021)	mRNA	2 doses at 21 days apart *.**	People aged 12 and over
Vaxzevria/Oxford–AstraZeneca (29 January 2021)	Viral vector	2 doses at 4 to 12 weeks apart **	People aged 18 and over
Janssen/Johnson&Johnson (11 March 2021)	Viral vector	1 dose **	People aged 18 and over

* For people with severe immunosuppression, 3 doses of the vaccine are recommended for the primary vaccination schedule (0, 21, 28 days for the Comirnaty/Pfizer–BioNTech vaccine, 0, 28, 28 days for the Spikevax/Moderna vaccine). Thus, in the case of severely immunosuppressed persons aged 12 years and over, the 3rd dose can be administered within the primary vaccination at least 28 days after the 2nd dose. Individuals falling within the following categories were considered severely immunocompromised persons and were recommended to receive three doses of the COVID-19 vaccine as their primary vaccination: patients with cancer in different stages of evolution with or without active treatment against it (chemotherapy, radiotherapy, molecular/biological treatment, etc.), patients with solid organ transplants or stem cell transplants with or without immunosuppressive treatment, and patients with congenital immunodeficiencies (DiGeorge syndrome, Wiskott–Aldrich syndrome, etc.) or with acquired immunodeficiencies (caused by HIV/AIDS or by a treatment). ** The booster dose could be administered at least 4 months after the primary vaccination (initially the recommendation was to administer each dose at least 6 months apart). A booster dose is especially recommended for people over 65 years of age, those with chronic conditions (regardless of age), people from medical and social centers, and those with a high risk of exposure (e.g., medical personnel). It is recommended an mRNA-based vaccine is used for the booster dose regardless of the vaccine used for the initial vaccination.

**Table 2 vaccines-11-00370-t002:** Distribution of doses by vaccine product, 27 December 2020–31 December 2021.

Vaccine Product	Vaxzevria AstraZeneca	Janssen Johnson& Johnson	Spikevax Moderna	Comirnaty Pfizer–BioNTech
Total doses distributed (%)	3,149,600 (11)	4,034,700 (14)	3,432,000 (12)	17,363,970 (62)
Total doses administered (%)	849,732 (5)	1,917,118 (12)	955,630 (6)	12,035,158 (76)
1st Dose (%)	431,860 (6)	1,903,921 (24)	403,611 (5)	5,143,066 (65)
2nd Dose (%)	417,859 (7)	13,124 (0)	398,470 (7)	5,049,245 (86)
3rd Dose (%)	13 (0)	73 (0)	153,549 (8)	1,842,847 (92)

**Table 3 vaccines-11-00370-t003:** Distribution of lost doses by vaccine product, 27 December 2020–31 December 2021.

Vaccine Product	Number of Lost Doses (% of Total Losses)	Rate of Loss from the Total Doses Distributed
Comirnaty/	426,866	2.5%
Pfizer–BioNTech	(61%)	
Spikevax/	110,679	3.2%
Moderna	(16%)	
Janssen/	99,586	2.0%
Johnson&Johnson	(14%)	
Vaxzevria/	63,244	2.5%
Astra Zeneca	(9%)	

## Data Availability

The datasets generated and analyzed during the current study are available from the corresponding author upon reasonable request.

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
