# Peer review of "COVID-19 Vaccination in Romania and the Benefits of the National Electronic Registry of Vaccinations"

_vaccines, 2023, doi:10.3390/vaccines11020370_

Round 1

Reviewer 1 Report

The authors have presented COVID-19 vaccine coverage and reach in Romania, while arguing that the national vaccine registry is essential. I have a number of suggestions:

1.     The entire paper has way too many paragraphs – reads a bit like a bulleted list of ideas. Could the authors weave these ideas together into paragraphs?

2.     Line 48 – why is the vaccine registry a necessity? This is an assumption that is made throughout the paper but is never really examined why the registry is valuable.

3.     Figure 2, would the authors use dates rather than week numbers?

4.     Although the authors state that they are assessing the national vaccine registry, the article is more descriptive of vaccine reach and coverage in Romania. I’m not seeing where the authors examine the utility and benefit of the vaccine registry.

Author Response

Thank you for the opportunity to improve our article.

Reviewer 1

Comments and Suggestions for Authors

The authors have presented COVID-19 vaccine coverage and reach in Romania, while arguing that the national vaccine registry is essential. I have a number of suggestions:

  1. The entire paper has way too many paragraphs – reads a bit like a bulleted list of ideas. Could the authors weave these ideas together into paragraphs?

Thank you for the suggestion, we have removed the subheadings with the necessary rephrasing.

  1. Line 48 – why is the vaccine registry a necessity? This is an assumption that is made throughout the paper but is never really examined why the registry is valuable.

In order to highlight the value of the registry we rephrased that line and added one clarifying paragraphs in the Discussion section:

“Since the beginning of the COVID-19 vaccination campaign in Romania, the National Registry of Vaccinations provided the information needed for the good development of the vaccination process (stock management in real time and vaccines supply accordingly, vaccines losses, vaccination coverage by population groups and territorial administrative unit). A part of this information has been transparently communicated to the Romanian population (on TV, social media, other websites).”

  1. Figure 2, would the authors use dates rather than week numbers?

Thank you, we changed to date format.

  1. Although the authors state that they are assessing the national vaccine registry, the article is more descriptive of vaccine reach and coverage in Romania. I’m not seeing where the authors examine the utility and benefit of the vaccine registry.

Thank you, we made some changes to highlight the importance of the vaccine registry.

Reviewer 2 Report

The manuscript entitled ‘COVID-19 vaccination in Romania and the benefits of National Electronic Registry of Vaccinations’ by Enciu et. al. summarizes the COVID19 vaccination data in Romania during 2020-2021 and emphasizes on importance of national documentation of this data. The authors highlight various points that could be useful output of the analysis of this vaccination data. This could be useful in drafting better vaccination policy for certain infections in future. The article is nicely written, and I have only couple of more points to add-

1-     All the vaccines might have not been available throughout the period (2020-2021). Some vaccines might have been introduced earlier than other ones. It will be a good idea to include this information as this will throw some light on why a particular vaccine was used more.

2-     One more piece of information - that authors can think about adding - is the cost involved with these vaccine candidates and if a certain population was charged for vaccination. May be specific county was charging. If this information is available, it will be worth including.  

3-     In Fig. 5, is there more vaccination in urban vs rural area as lack of vaccine delivery services may make this more complicated.

Author Response

Thank you for the opportunity to improve our article.

Reviewer 2

Comments and Suggestions for Authors

The manuscript entitled ‘COVID-19 vaccination in Romania and the benefits of National Electronic Registry of Vaccinations’ by Enciu et. al. summarizes the COVID19 vaccination data in Romania during 2020-2021 and emphasizes on importance of national documentation of this data. The authors highlight various points that could be useful output of the analysis of this vaccination data. This could be useful in drafting better vaccination policy for certain infections in future. The article is nicely written, and I have only couple of more points to add-

1-     All the vaccines might have not been available throughout the period (2020-2021). Some vaccines might have been introduced earlier than other ones. It will be a good idea to include this information as this will throw some light on why a particular vaccine was used more.

Thank you, we have added the information suggested as it follows:

“The first available vaccine was Comirnaty/Pfizer-BioNTech vaccine. The other three vaccines became available in Romania in the following order: Spikevax/Moderna (first used in Romania on 4 February 2021), Oxford/Astra-Zeneca (first used in Romania on 15 February 2021) and Janssen/Johnson&Johnson (first used in Romania on 04 May 2021) “.

2-     One more piece of information - that authors can think about adding - is the cost involved with these vaccine candidates and if a certain population was charged for vaccination. May be specific county was charging. If this information is available, it will be worth including. 

Thank you for the comment. In Romania, the COVID-19 vaccination is voluntary and free of charge, and we clarified this also in the manuscript. The cost of the vaccination is covered by the authorities (Ministry of Health).

3-     In Fig. 5, is there more vaccination in urban vs rural area as lack of vaccine delivery services may make this more complicated.

We have replaced this figure with a more explicit one presenting the vaccination coverage by administrative territorial unit (locality, city, county), generated by the National Vaccination Registry.

In order to ensure the equitable access to COVID-19 vaccines in rural and remote areas, the COVID-19 vaccination strategy included the organization of mobile vaccination teams.  

Reviewer 3 Report

The authors have made an interesting attempt at “COVID-19 vaccination in Romania and the benefits of National Electronic Registry of Vaccinations.” The manuscript is interesting; however, the authors need to justify the scientific writing manuscript. Some of the general comments are provided below:

1.     Were all vaccines available at the same time?

2.     Can the authors explain the reasons for the loss of vaccines?

3.     What are the different side effects of each vaccine?

4.     Are the investigators confident and even certain, that they were able to gain access to all available data?

5.     Further, how did the investigators gain access to these online reviews? Were they publicly accessible or were the investigators required to obtain permission to gain access? In addition, what is involved in the data collection (ie, ‘scraping’) process?

6.     The article would also be strengthened if the authors included some description of what was learned from assessments of previous mass vaccination clinics and compare that with what they learned in their assessment.

7.     It would be interesting if authors can discuss the death rate among COVID-19 vaccinated and non-vaccinated individuals.

Author Response

Thank you for the opportunity to improve our article.

Reviewer 3

Comments and Suggestions for Authors

The authors have made an interesting attempt at “COVID-19 vaccination in Romania and the benefits of National Electronic Registry of Vaccinations.” The manuscript is interesting; however, the authors need to justify the scientific writing manuscript. Some of the general comments are provided below:

  1. Were all vaccines available at the same time?

Thank you for the question, the vaccines became available as they were authorized by European Medicine Agency, not at the same time. We have added this information in the manuscript.  

“The first available vaccine was Comirnaty/Pfizer-BioNTech vaccine and the vaccination campaign started on 27 December 2020. The other three vaccines became available in Romania in the following order: Spikevax/Moderna (first used in Romania on 4 February 2021), Oxford/Astra-Zeneca (first used in Romania on 15 February 2021) and Janssen/Johnson&Johnson (first used in Romania on 04 May 2021) “.

  1. Can the authors explain the reasons for the loss of vaccines?

Thank you for the question, we added the explanation in the manuscript.

The reasons for loss of vaccines are especially related to the fact that the COVID-19 vaccines are conditioned in multi-dose vials (and for the booster dose with Spikevax vaccine was used half a dose) and with limited stability after dilution. To reduce the vaccines losses at the beginning of the vaccination campaign when the availability of vaccines was limited, the people who want to get vaccinated were scheduled for vaccination. As the availability of vaccines increased, in order to obtain a higher vaccination coverage, the vaccination was possible without appointment.

  1. What are the different side effects of each vaccine?

In this article, our aim is to present the utility of National Registry of Vaccinations in providing information useful for improving vaccination strategy in general. We plan a further work on the safety of COVID-19 vaccines.

  1. Are the investigators confident and even certain, that they were able to gain access to all available data?

The accuracy and completeness of data is dependent on the users who record the vaccinations, but we are confident that some facts such as the need of the vaccination proof (obtained only by recording the vaccination performed in the National Electronic Registry of Vaccinations, the legal framework and the users’ training) have contributed at creating a database at high level of completeness and accuracy.

  1. Further, how did the investigators gain access to these online reviews? Were they publicly accessible or were the investigators required to obtain permission to gain access? In addition, what is involved in the data collection (ie, ‘scraping’) process?

Thank you, we added a clarification in the Method section.

The National Centre for Communicable Diseases Surveillance and Control is responsible for the management of the National Electronic Registry of Vaccinations data, the authors with right of access contributing at the development, implementation, and management of this database.

Furthermore, a part of this data is publicly available on https://vaccinare-covid.gov.ro/noutati/ and on the National Institute of Public Health website (https://insp.gov.ro/centrul-national-de-supraveghere-si-control-al-bolilor-transmisibile-cnscbt/infectia-cu-noul-coronavirus-sars-cov-2/raportare-saptamanala-vaccinare-impotriva-covid-19/).

  1. The article would also be strengthened if the authors included some description of what was learned from assessments of previous mass vaccination clinics and compare that with what they learned in their assessment.

Thank you for the comment. As we previously mentioned, the aim of this article is to present only the benefits of the National Electronic Registry of Vaccinations in providing information useful for improving the vaccination strategy in real time.

  1. It would be interesting if authors can discuss the death rate among COVID-19 vaccinated and non-vaccinated individuals.

Thank you, this is a good point. We plan to analyse such characteristics of COVID-19 vaccination process in a further work.

Round 2

Reviewer 1 Report

The authors have addressed most of my concerns. Throughout the article would benefit by reducing the number of paragraphs. (Each sentence is its own paragraph).

Author Response

Thank you for the suggestion, we have reduced the number of paragraphs.

Reviewer 3 Report

The authors have modified the manuscript,  now it is acceptable for the publication. 

Author Response

Thank you for the revision of our manuscript.